# We are Who We Cite: Bridges of Influence Between Natural Language Processing and Other Academic Fields

**Jan Philip Wahle**
National Research Council Canada
University of Göttingen Germany
wahle@uni-goettingen.de

**Terry Ruas**
University of Göttingen
Göttingen, Germany
ruas@uni-goettingen.de

**Mohamed Abdalla**
Institute for Better Health
Toronto, Canada
msa@cs.toronto.edu

**Bela Gipp**
University of Göttingen
Göttingen, Germany
gipp@uni-goettingen.de

**Saif M. Mohammad**
National Research Council Canada
Ottawa, Canada
saif.mohammad@nrc-cnrc.gc.ca

## Abstract

Natural Language Processing (NLP) is poised to substantially influence the world. However, significant progress comes hand-in-hand with substantial risks. Addressing them requires broad engagement with various fields of study. Yet, little empirical work examines the state of such engagement (past or current). In this paper, we quantify the degree of influence between 23 fields of study and NLP (on each other). We analyzed ~77k NLP papers, ~3.1m citations from NLP papers to other papers, and ~1.8m citations from other papers to NLP papers. We show that, unlike most fields, the cross-field engagement of NLP, measured by our proposed *Citation Field Diversity Index (CFDI)*, has declined from 0.58 in 1980 to 0.31 in 2022 (an all-time low). In addition, we find that NLP has grown more insular—citing increasingly more NLP papers and having fewer papers that act as bridges between fields. NLP citations are dominated by computer science; Less than 8% of NLP citations are to linguistics, and less than 3% are to math and psychology. These findings underscore NLP's urgent need to reflect on its engagement with various fields.

## 1 Introduction

The degree to which the ideas and artifacts of a field of study are useful to the world (for improving our understanding of the world or for practical applications) is a measure of its influence. Developing a better sense of the influence of a field has several benefits, such as understanding what fosters greater innovation and what stifles it; what a field has success at understanding and what remains elusive; who are the most prominent stakeholders benefiting and who are being left behind, etc. However, the modes of influence are numerous and complex; thus, making an empirical determination of the degree of influence difficult.

In this work, we focus on a specific subset of influence: the scientific influence of one field of study on another. Stated another way, we explore the degree to which the ideas and scientific artifacts of various fields of study impact a target field of interest and the degree to which the ideas of a target field impact other fields.

> *Who is influencing us (our field)?*
> *Who are we influencing?*

Mechanisms of field-to-field influence are also complex, but one notable marker of scientific influence is citations. Thus, we propose that the extent to which a source field cites a target field is a rough indicator of the degree of influence of the target on the source. We note here, though, that not all citations are equal—some cited work may have been much more influential to the citing work than others (Zhu et al., 2015; Valenzuela et al., 2015). Further, citation patterns are subject to various biases (Mohammad, 2020a; Ioannidis et al., 2019). Nonetheless, meaningful inferences can be drawn at an aggregate level; for example, if the proportion of citations from field *x* to a target field *y* has markedly increased as compared to the proportion of citations from other fields to the target, then it is likely that the influence of *x* on *y* has grown.

*Agency of the Researcher*: A key point about scientific influence is that as researchers, we are not just passive objects of influence. Whereas some influences are hard to ignore (say, because of substantial buy-in from one's peers and the review process), others are a direct result of active engagement by the researcher with relevant literature (possibly from a different field). We

(can) choose whose literature to engage with, and thus benefit from. Similarly, while at times we may happen to influence other fields, we (can) choose to engage with other fields so that they are more aware of our work. Such an engagement can be through cross-field exchanges at conferences, blog posts about one's work for a target audience in another field, etc. Thus, examining which fields of study are the prominent sources of ideas for a field is a window into what literature the field as a whole values, engages with, and benefits from.

> *Whose literature are we (our field)*
> *choosing to engage with?*
> *Who is engaging with our literature?*

Further, since cross-field engagement leads to citations, we argue:

> *Who we cite says a lot about us.* And,
> *Who cites us also says a lot about us.*

*Why NLP*: While studying influence is useful for any field of study, and our general approach and experimental setup are broadly applicable, we focus on Natural language Processing (NLP) research for one critical reason.

NLP is at an inflection point. Recent developments in large language models have captured the imagination of the scientific world, industry, and the general public. Thus, NLP is poised to exert substantial influence, despite significant risks (Buolamwini and Gebru, 2018; Mitchell et al., 2019; Bender et al., 2023; Mohammad, 2022; Wahle et al., 2023; Abdalla et al., 2023). Further, language is social, and NLP applications have complex social implications. Therefore, responsible research and development need engagement with a wide swathe of literature (arguably, more so for NLP than other fields). Yet, there is no empirical study on the bridges of influence between NLP and other fields.

Additionally, even though NLP is interdisciplinary and draws on many academic disciplines, including linguistics, computer science (CS), and artificial intelligence (AI), CS methods and ideas dominate the field. Relatedly, according to the '*NLP Community Metasurvey*', most respondents believed that NLP should do more to incorporate insights from relevant fields, despite believing that the community did not care about increasing interdisciplinarity (Michael et al., 2022). Thus, we believe it is important to measure the extent to which CS and non-CS fields influence NLP.

In this paper, we describe how we created a new dataset of metadata associated with ∼77k NLP papers, ∼3.1m citations from NLP papers to other papers, and ∼1.8m citations from other papers to NLP papers. Notably, the metadata includes the field of study and year of publication of the papers *cited by* the NLP papers, the field of study and year of publication of papers *citing* NLP papers, and the NLP subfields relevant to each NLP paper. We trace hundreds of thousands of citations in the dataset to systematically and quantitatively examine broad trends in the influence of various fields of study on NLP and NLP's influence on them. Specifically, we explore:

1. Which fields of study are influencing NLP? And to what degree?

2. Which fields of study are influenced by NLP? And to what degree?

3. To what extent is NLP insular—building on its own ideas as opposed to being a hotbed of cross-pollination by actively bringing in ideas and innovations from other fields of study?

4. How have the above evolved over time?

This study enables a broader reflection on the extent to which we are actively engaging with the communities and literature from other fields—an important facet of responsible NLP.

## 2 Related Work

Previous studies on responsible research in NLP have examined aspects such as author gender diversity (Vogel and Jurafsky, 2012; Schluter, 2018; Mohammad, 2020b), author location diversity (Rungta et al., 2022), temporal citation diversity (Singh et al., 2023), software package diversity (Gururaja et al., 2023), and institutional diversity (Abdalla et al., 2023). However, a core component of responsible research is interdisciplinary engagement with other research fields (Porter et al., 2008; Leydesdorff et al., 2019). Such engagement can steer the direction of research. Cross-field research engagement can provide many benefits in the short-term, by integrating expertise from various disciplines to address a specific issue, and in the long-term to generate new insights or innovations by blending ideas from different disciplines (Hansson, 1999).

Many significant advances have emerged from the synergistic interaction of multiple disciplines, e.g., the conception of quantum mechanics, a theory that coalesced Planck's idea of quantized energy levels (Planck, 1901), Einstein's photoelectric effect (Einstein, 1905), and Bohr's model of the

atom (Bohr, 1913). In addition, entirely new fields have emerged such as ecological economics. A key concept in ecological economics is the notion of 'ecosystem services', which originated in biology (Postel et al., 2012) and ecology (Pearce and Turner, 1989). The field of medicine has integrated neuroscience (Bear et al., 2020) with engineering principles (Saltzman, 2009). Within NLP, in the 1980s, there was a marked growth in the use of statistical probability tools to process speech, driven by researchers with electrical engineering backgrounds engaging with NLP (Jurafsky, 2016).

To quantify the interdisciplinary of a body of research, researchers have proposed various metrics (on Science et al., 2004; Wang et al., 2015). These metrics can range from simple counting of referenced subjects (Wang et al., 2015) to more complex metrics such as the Gini-index (Leydesdorff and Rafols, 2011). Our work seeks to understand the level of interdisciplinarity in NLP by tracking how often and how much it references other fields and how this pattern evolves over time.

Several studies have reported an increasing trend of interdisciplinarity across fields (Porter and Rafols, 2009; Larivière et al., 2009; Van Noorden et al., 2015; Truc et al., 2020) and within subfields (Pan et al., 2012). For those observing increasing interdisciplinarity, the growth is predominantly seen in neighboring fields, with minimal increases in connections between traditionally distant research areas, such as materials sciences and clinical medicine (Porter and Rafols, 2009). The findings are contested, with other researchers finding that the "*scope of science*" has become more limited (Evans, 2008), or that the findings are field-dependent, exemplified by contrasting citation patterns in sub-disciplines within anthropology and physics (Choi, 1988; Pan et al., 2012).

Within computer science (CS), Chakraborty (2018) used citational analysis to show increasing interdisciplinarity in CS sub-fields, positing a three-stage life cycle for citation patterns of fields: a growing, maturing, and interdisciplinary phase.

Within NLP, Anderson et al. (2012) examined papers from 1980 to 2008 to track the popularity of research topics, the influence of subfields on each other, and the influence of government interventions on the influx of new researchers into NLP. By examining the trends of NLP's engagement with other fields, our work seeks to provide a discipline-specific perspective.

| Time range | 1965–2022 |
|---|---|
| #papers | 209 016 314 |
| #citations | 2 521 776 124 |
| #papers NLP | 76 745 |
| #out-citations from NLP | 3 115 126 |
| #in-citations to NLP | 1 847 873 |

Table 1: Overall dataset statistics.

## 3 Data

Central to work on field-to-field influence is a dataset where the field of each of the papers is known. We would like to note here that a paper may belong to many fields, each to varying extents, and it can be difficult for humans and automatic systems to determine these labels and scores perfectly. Further, it is difficult to obtain a complete set of papers pertaining to each field; even determining what constitutes a field and what does not is complicated. Nonetheless, meaningful inferences can be drawn from large samples of papers labeled as per some schema of field labels.

As in past work (Mohammad, 2020a; Abdalla et al., 2023), we chose the *ACL Anthology (AA)* as a suitable sample of NLP papers.[1] We extracted metadata associated with the NLP (AA) papers and with papers from various fields using a Semantic Scholar (S2) dump.[2] It includes papers published between 1965–2022, totaling 209m papers and 2.5b citations. S2's field of study annotation labels each paper as belonging to one or more of 23 broad fields of study. It uses a field-of-study classifier based on abstracts and titles (S2FOS[3]) with an accuracy of 86%. The S2 dump also includes information about which paper cites another paper. Table 1 provides an overview of key dataset statistics.

Since we expect NLP papers to have cited CS substantially, we also examine the influence between specific subfields of CS and NLP. Thus, we make use of the subfield annotations of the *Computer Science Ontology (CSO)* version 3.3 (Salatino et al., 2020) that is available through the *D3 dataset* (Wahle et al., 2022). The CSO classifier uses a paper's titles, abstracts, and keywords and has an f1-score of 74% (Salatino et al., 2019).

Within the CSO ontology, we consider only the 15 most populous subfields of CS (e.g., AI, computer vision (CV)). We also further split AI into Machine Learning (ML) and the rest of AI (AI') — simply because ML is a known influencer of NLP.

---

[1] https://www.aclweb.org/anthology
[2] https://api.semanticscholar.org/api-docs/datasets
[3] https://tinyurl.com/8a7anf2a

The source code to process our data and reproduce the experiments is available on GitHub:

https://github.com/jpwahle/
emnlp23-citation-field-influence

## 4 Experiments

We use the dataset described above to answer a series of questions about the degree of influence between various fields and NLP.

**Q1.** *Which fields do NLP papers cite? In other words, how diverse are the outgoing citations by NLP papers—do we cite papers mostly just from computer science, or do we have a marked number of citations to other fields of study, as well?*

*Separately, which fields cite NLP papers? How diverse are the incoming citations to NLP papers?*

**Ans.** For each NLP paper, we look at the citations to other papers and count the number of outgoing citations going to each field (or, *outcites*). Similarly, we track papers citing NLP work and count the number of incoming citations from each field (or, *incites*). If a paper is in multiple fields, we assign one citation to each field.

We calculate the percentage of outgoing citations: the percent of citations that go to a specific field from NLP over all citations from NLP to any field. Similarly, we calculate the percentage for incoming citations: the percent of citations going from a specific field to NLP over all citations from any field to NLP. Because we expect CS to be a marked driver of NLP citations, we split citations into CS and non-CS and then explore citations within CS and non-CS for additional insights.

*Results.* Figure 1 shows three Sankey plots. The right side of each figure shows citations from other fields to NLP (*#incoming citations, %incomcing citations*), and the left side shows citations from NLP to other fields (*#outgoing citations, %outgoing citations*). The middle part shows the overall number of outgoing and incoming citations from/to NLP (*#outgoing citations, #incoming citations*). The width of the grey flowpath is proportional to the number of citations.

Figure 1 (a) shows the flow of citations from CS and non-CS papers to NLP (right) and from NLP to CS and non-CS papers (left). Overall, the number of incoming citations to NLP is lower (1.9m) than the number of outgoing citations (3.1m). The discrepancy between incoming and outgoing citations shows that NLP is overall cited less than it cites

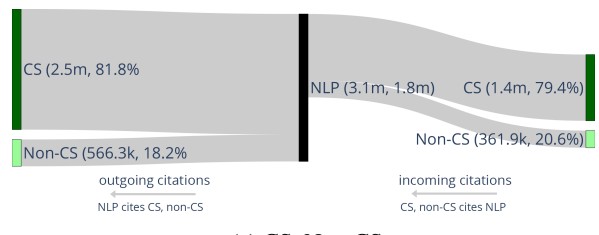

(a) CS, Non-CS

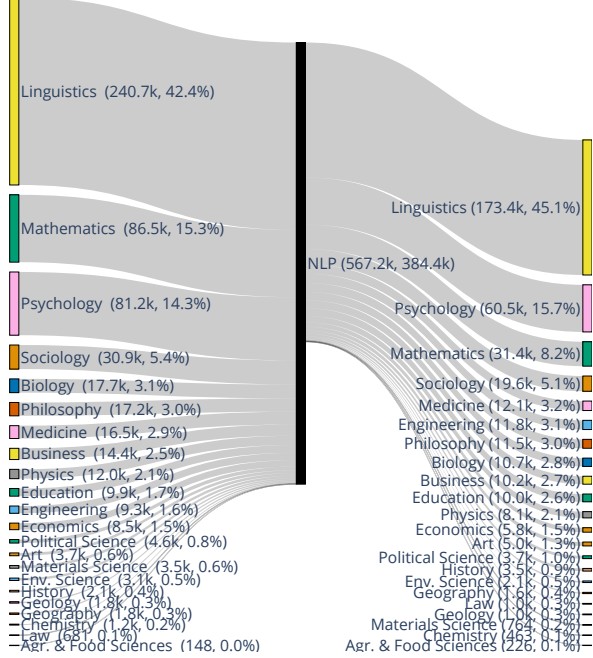

(b) Non-CS Fields

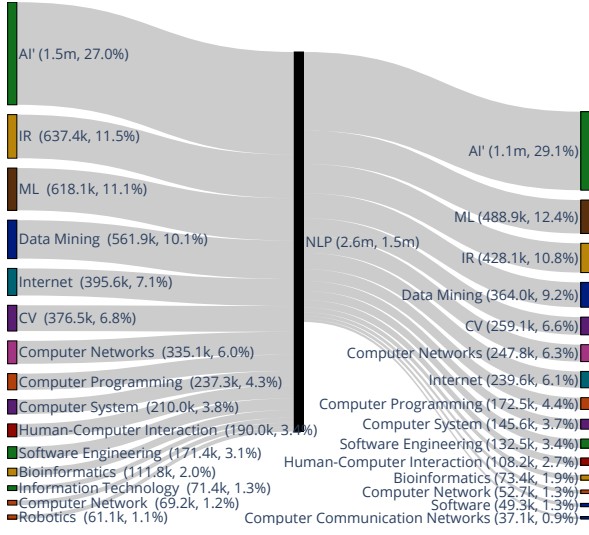

(c) CS Subfields

Figure 1: Citations from other fields to NLP (right) and from NLP to other fields (left).

other fields. This is not uncommon: we found that *#incites* is lower compared to *#outcites* for all 23 fields of study (figures not shown here).

79.4% of citations received by NLP are from CS papers. Similarly, a large majority (81.8%) of citations from NLP papers go to CS papers. Citations

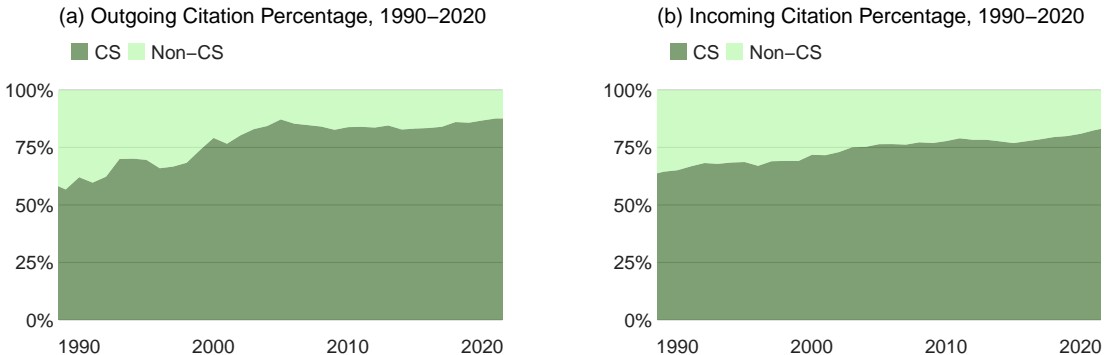

Figure 2: The percentage of citations (a) from NLP to CS and non-CS and (b) from CS and non-CS to NLP over all citations from and to NLP with a moving average of three years.

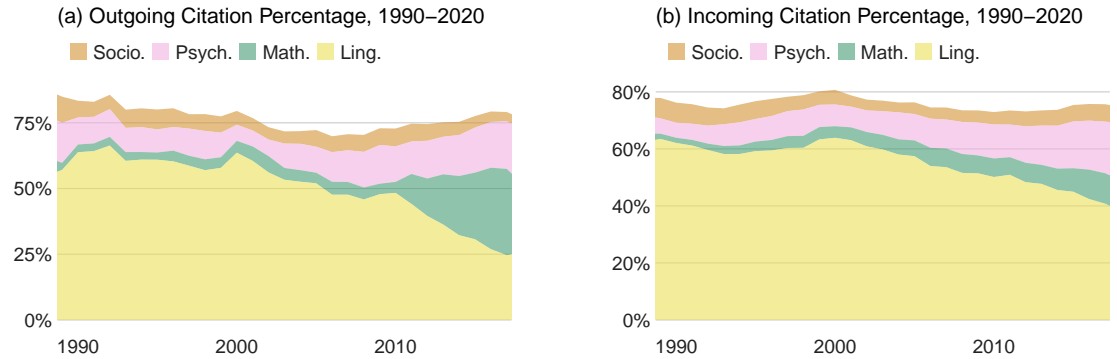

Figure 3: The percentage of citations (a) from NLP to non-CS fields and (b) non-CS fields to NLP in relation to all non-CS citations from and to NLP.

to individual non-CS fields are markedly lower than CS: linguistics (7.6%), math (2.8%), psychology (2.6%), and sociology (1.0%) (Figure 13 in the Appendix shows the percentages for all 23 fields).

To get a better sense of the variation across non-CS fields, Figure 1 (b) shows a Sankey plot when we only consider the non-CS fields. Observe that linguistics receives 42.4% of all citations from NLP to non-CS papers, and 45.1% of citations from non-CS papers to NLP are from linguistics. NLP cites mathematics more than psychology, but more of NLP's incites are from psychology than math.

Figure 1 (c) shows the flow of citations from CS subfields to NLP and from NLP to CS subfields. Observe that NLP cites the following subfields most: AI' (27.0%), ML (11.1%), and information retrieval (IR) (11.5%). ML receives roughly as many citations from NLP as all non-CS fields combined (2.5x #citations as linguistics).

*Discussion.* Even though we expected a high CS–NLP citational influence, this work shows for the first time that ∼80% of the citations in and out of NLP are from and to CS. Linguistics, mathematics, psychology, and social science are involved in a major portion of the remaining in- and out-citations.

However, these results are for all citations and do not tell us whether the citational influences of individual fields are rising or dropping.

**Q2.** *How have NLP's outgoing (and incoming) citations to fields changed over time?*

**Ans.** Figure 2 shows NLP's outgoing (a) and incoming (b) citation percentages to CS and non-CS papers from 1990 to 2020. Observe that while in 1990 only about 54% of the outgoing citations were to CS, that number has steadily increased and reached 83% by 2020. This is a dramatic transformation and shows how CS-centric NLP has become over the years. The plot for incoming citations (Figure 2 (b)) shows that NLP receives most citations from CS, and that has also increased steadily from about 64% to about 81% in 2020.

Figure 3 shows the citation percentages for four non-CS fields with the most citations over time. Observe that linguistics has experienced a marked (relative) decline in relevance for NLP from 2000 to 2020 (60.3% to 26.9% for outcites; 62.7% to 39.6% for incites). Details on the diachronic trends for CS subfields can be found in the Appendix A.3.

*Discussion.* Over time, both the in- and out-citations of CS have had an increasing trend. These

results also show a waning influence of linguistics and sociology and an exponential rise in the influence of mathematics (probably due to the increasing dominance of mathematics-heavy deep learning and large language models) and psychology (probably due to increased use of psychological models of behavior, emotion, and well-being in NLP applications). The large increase in the influence of mathematics seems to have largely eaten into what used to be the influence of linguistics.

**Q3.** *Which fields cite NLP more than average and which do not? Which fields are cited by NLP more than average, and which are not?*

**Ans.** As we know from Q1, 15.3% of NLP's non-CS citations go to math, but how does that compare to other fields citing math? Are we citing math more prominently than the average paper? That is the motivation behind exploring this question. To answer it, we calculate the difference between NLP's outgoing citation percentage to a field $f$ and the macro average of the outgoing citations from various fields to $f$. We name this metric *Outgoing Relative Citational Prominence (ORCP)*. If NLP has an ORCP greater than 0 for $f$, then NLP cites $f$ more often than the average of all fields to $f$.

$$\text{ORCP}_{NLP}(f) = X(f) - Y(f) \quad (1)$$

$$X(f) = \frac{C^{NLP \to f}}{\sum_{\forall f_j \in F} C^{NLP \to f_j}} \quad (2)$$

$$Y(f) = \frac{1}{N} \sum_{i=1}^{N} \frac{C^{f_i \to f}}{\sum_{\forall f_j \in F} C^{f_i \to f_j}} \quad (3)$$

where $F$ is the set of all fields, $N$ is the number of all fields, and $C^{f_i \to f_j}$ is the number of citations from field $f_i$ to field $f_j$.

Similar to ORCP, we calculate the *Incoming Relative Citational Prominence (IRCP)* from each field to NLP. IRCP indicates whether a field cites NLP more often than the average of all fields to NLP.

*Results.* Figure 4 shows a plot of NLP's ORCPs with various fields. Observe that NLP cites CS markedly more than average (ORCP = 73.9%). Even though linguistics is the primary source for theories of languages, NLP papers cite only 6.3 percentage points more than average (markedly lower than CS). It is interesting to note that even though psychology is the third most cited non-CS field by NLP (see Figure 1 (b)), it has an ORCP of $-5.0$, indicating that NLP cites psychology markedly less than how much the other fields cite psychology.

Figure 14 in the Appendix is a similar plot as Figure 4, but also shows ORCP scores for CS, psy-

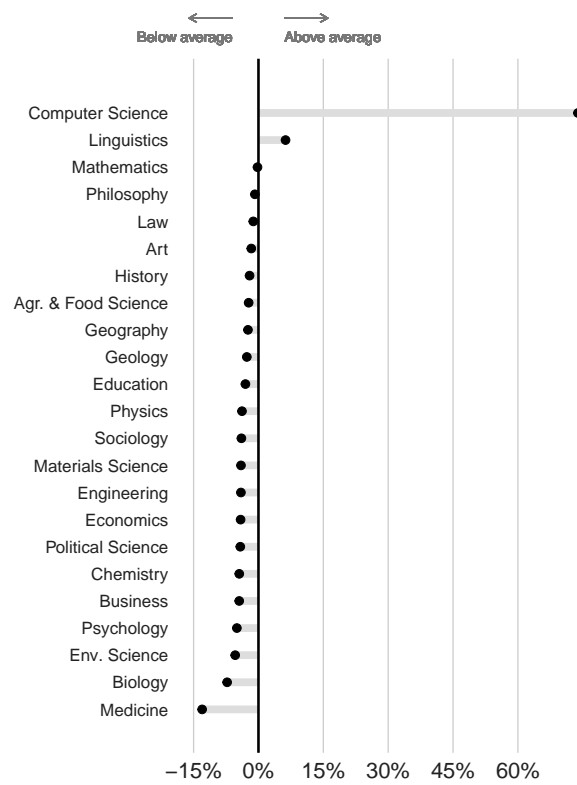

Figure 4: NLP's Outgoing Relative Citational Prominence (ORCP) scores for 23 fields of study.

chology, linguistics, and math. Here, the gray horizontal bands indicate the range from min. to max. of all 23 fields of study. Among the 23 fields, the highest ORCP to a field is reached by the field itself, showing that, as expected, citations to papers within the same field are higher than cross-field citations. However, none of the 23 fields cite CS as much as NLP: NLP cites CS 73.9 percentage points more than average, whereas CS cites itself only 32.6 percentage points more than average. Linguistics cites psychology substantially above average with 15% ORCP and CS markedly too with 6.8% ORCP. Psychology has a strong medicine focus citing it more than 16 percentage points above average. Math cites physics and engineering above average but not nearly as much as CS.

Figures 15 and 16 (Appendix) show plots for incoming RCP. Similar trends are observed as seen for outgoing RCP.

*Discussion.* Overall, the analyses show a dramatic degree of CS-centric NLP citations.

**Q4.** *Is there a bottom-line metric that captures the degree of diversity of outgoing citations? How did the diversity of outgoing citations to NLP papers (as captured by this metric) change over time? And the same questions for incoming citations?*

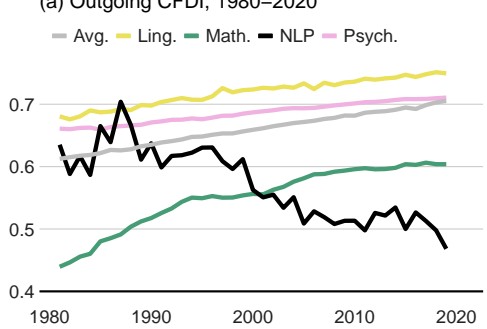

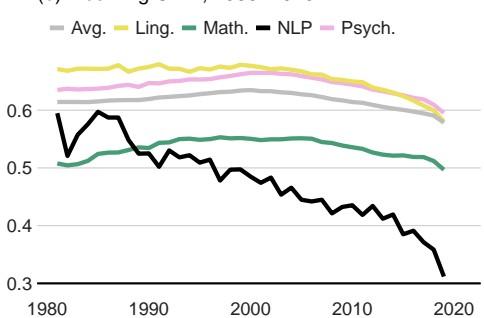

Figure 5: CFDI of NLP and the three largest fields cited by NLP for (a) outgoing citations and (b) incoming citations of that field. The macro-average shows CFDI for the average over all 23 fields.

**Ans.** To measure citational field diversity in a single score, we propose the *Citation Field Diversity Index (CFDI)*. Outgoing CFDI is defined as:

$$\text{CFDI} = 1 - \sum_{f \in fields} p_f^2 \qquad (4)$$
$$\text{where } p_f = x_f / X, \qquad (5)$$
$$\text{and } X = \sum_i^N x_i \qquad (6)$$

where $x_f$ is the number of papers in field $f$, and $N$ is the total number of citations. CFDI has a range of $[0, 1]$. Scores close to 1 indicate that the number of citations from the target field (in our case, NLP) to each of the 23 fields is roughly uniform. A score of 0 indicates that all the citations go to just one field. Incoming CFDI is calculated similarly, except by considering citations from other fields to the target field (NLP).

*Results.* The overall outgoing CFDI for NLP papers is 0.57, and the overall incoming CFDI is 0.48. Given the CS dominance in citations observed earlier in the paper, it is not surprising that these scores are closer to 0.5 than to 1.

Figure 5 (a) shows outgoing CFDI over time for NLP papers. It also shows CFDIs for linguistics, math, and psychology (the top 3 non-CS fields citing and cited by NLP), as well as the macro average of the CFDIs of all fields. Observe that while the average outgoing CFDI has been slowly increasing with time, NLP has experienced a swift decline over the last four decades. The average outgoing CFDI has grown by 0.08 while NLP's outgoing CFDI declined by about 30% (0.18) from 1980–2020. CFDI for linguistics and psychology follow a similar trend to the average while math has experienced a large increase in outgoing CFDI of 0.16 over four decades.

Similar to outgoing CFDI, NLP papers also have a marked decline in incoming CFDI over time; in-

dicating that incoming citations are coming largely from one (or few fields) as opposed to many. Figure 5 (b) shows the plot. While the incoming CFDIs for other fields, as well as the average, have been plateauing from 1980 to 2010, NLP's incoming CFDI has decreased from 0.59 to 0.42. In the 2010s, all fields declined in CFDI, but NLP had a particularly strong fall (0.42 to 0.31).

*Discussion.* The decline in both incoming and outgoing field diversity within the NLP domain indicates significantly less cross-field engagement and reliance on existing NLP / CS research. In contrast, other large fields like math and psychology have maintained or increased CFDI scores. NLP was a much smaller field in the 80s and 90s compared to more established fields like psychology or math; and NLP has had a marked trajectory of growth that may have been one of the driving forces of the observed trends. Nonetheless, this stark contrast between NLP and other fields should give the NLP community pause and ask whether we are engaging enough with relevant literature from various fields.

**Q5.** *Given that a paper can belong to one or more fields, on average, what is the number of fields that an NLP paper belongs to (how interdisciplinary is NLP)? How does that compare to other academic fields? How has that changed over time?*

**Ans.** We determine the number of field labels each NLP paper has and compare it to the average for papers from other fields of study.

*Results.* Figure 6 shows that the average number of fields per NLP paper was comparable to that of other fields in 1980. However, the trends for NLP and other fields from 1980 to 2020 diverge sharply. While papers in other fields have become slightly more interdisciplinary, a finding that is consistent with related work (Porter and Rafols,

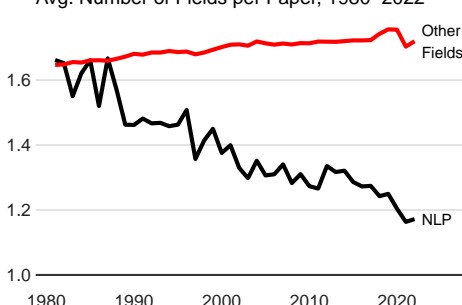

Figure 6: Avg. number of fields per paper.

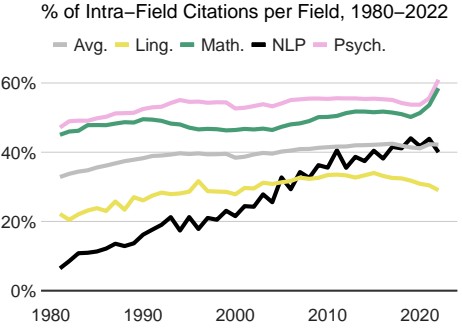

Figure 7: Intra-field citation percentage of fields.

2009; Van Noorden et al., 2015; Truc et al., 2020), NLP papers have become, on average, less concerned with multiple fields.

*Discussion.* NLP papers demonstrate a decreasing trend in interdisciplinarity, in contrast to other fields that show increased heterogeneity in their publications. This decline in NLP possibly reflects the field's increased specialization and focus on computational approaches, at the expense of work that holistically situates NLP systems in society.

**Q6.** *To what extent do NLP papers cite other NLP papers as opposed to outside-NLP papers?*

**Ans.** For NLP and each of the 23 fields of study, we measure the percentage of intra-field citations i.e., the number of citations from a field to itself over citations to all fields (including itself).

*Results.* Figure 7 shows the percentage of intra-field citations for NLP, linguistics, math, psychology, and the macro average of all 23 fields. In 1980, 5%, or every 20th citation from an NLP paper, was to another NLP paper. Since then, the proportion has increased substantially, to 20% in 2000 and 40% in 2020, a trend of 10% per decade. In 2022, NLP reached the same intra-field citation percentage as the average intra-field citation percentage over all fields. 40% is still a lower bound as we use the AA to assign a paper to NLP. This percentage might be higher when relaxing this condition to papers outside the AA. Compared to other fields, such as linguistics, NLP experienced particularly strong growth in intra-field citations after a lower score start. This is probably because NLP, as field, is younger than others and was much smaller during the 1980s and 90s. Linguistics's intra-field citation percentage has increased slowly from 21% in 1980 to 33% in 2010 but decreased ever since. Math and psychology had a plateauing percentage over time but recently, experienced a swift increase again.

*Discussion.* The rise of within-field citations in NLP signifies increasing insularity, potentially due to its methodological specialization. It is still unclear why NLP is citing itself proportionally more than other fields. One can argue that as the body of work in NLP grows, there is plenty to cite in NLP itself; perhaps the cited work itself cites outside work, but authors of current work do not feel the need to cite original work published in outside fields, as much as before. Yet, it is not clear why this occurs only in NLP and not in other fields (which are also growing).

**Q7.** *Do well-cited NLP papers have higher citational field diversity?*

**Ans.** We measure outgoing CFDI for each paper in nine different citation bins to distinguish highly cited papers from low cited papers: 0, 1-9, 10-49, 50-99, 100-499, 500-999, 1000-1999, 2000-4999, and 5000+ citations. To track trends of outgoing citational field diversity over time, we group papers in four time ranges: 1965–1990, 1990–2000, 2000–2010, and 2010–2020.

*Results.* Figure 8 shows the results. Observe that for both 1965–1990 and 1990–2000, higher cited papers have higher outgoing CFDI (with few exceptions). From 2000–2010, the overall outgoing CFDI is lower, as seen before in Figure 5 (a), but the difference in CFDI between citation groups has plateaued. The trend then reverses between 2010–2020, and higher cited papers have less outgoing citational field diversity.

*Discussion.* Papers may garner citations for various reasons, and those that get large amounts of citations are not necessarily model papers (or perfect in every way). However, by virtue of their visibility, highly-cited papers markedly impact research and how early researchers perceive papers should be written. There have been concerns in the academic

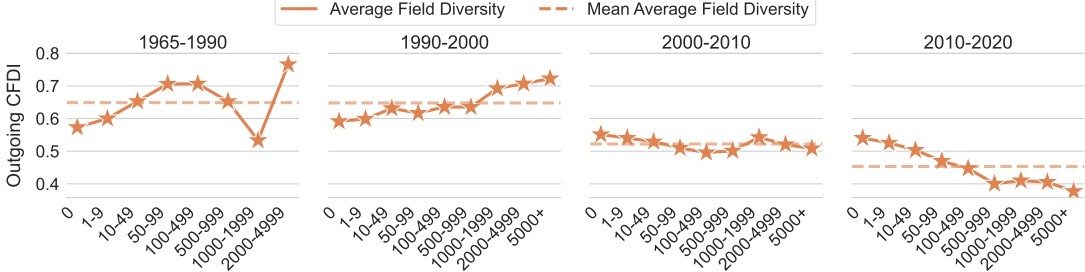

Figure 8: Outgoing citational field diversity for NLP papers in different citation groups over time.

community that early-stage researchers often categorize papers not from their immediate field of study as irrelevant. This coincides with a marked increase in papers posted on arXiv, and researchers feel that it is difficult to keep track of core NLP papers itself. Thus, engaging with literature from outside of NLP and CS may be getting sidelined even further. This recent trend of highly cited papers not being as diverse in their engagement with literature as those from earlier decades is alarming.

**Q8.** *Is there an online tool that allows one to easily determine the diversity of fields being cited by a paper (or a set of papers)?*

**Ans.** We have developed a freely accessible web-based tool to promote cognizance of disciplinary diversity in academic citations.[4] Users can upload a paper's PDF, input an ACL Anthology or Semantic Scholar link (including author profiles or proceedings), and the system produces salient data and visualizations concerning the diversity of fields of the cited literature. Details in Appendix A.1.

## 5 Concluding Remarks

In this work, we examined the citational influence between NLP and various fields of study. We created a distinct dataset of metadata that includes ∼77k NLP papers, citations to these papers (and the fields the citing papers belong to), and citations by the NLP papers (and the fields the cited papers belong to). We analyzed this data using various metrics such as *Citational Field Diversity Index* and *Relative Citational Prominence* to show that, unlike most other fields, the diversity of both incoming and outgoing citations to and from NLP papers has steadily declined over the past few decades.

Our experiments show that NLP citations are dominated by CS, accounting for over 80% of citations, with particular emphasis on AI, ML, and IR. Contributions from non-CS fields like linguis-

tics and sociology have diminished over time. The diversity of NLP citing papers from other fields and how NLP is cited by other fields has decreased since 1980 and is particularly low compared to other fields such as linguistics and psychology. NLP presents increasing insularity reflected by the growth of intra-field citations and a decline in multi-disciplinary works. Although historically, well-cited papers have higher citational field diversity, concerningly, this trend has reversed in the 2010s.

Over the last 5 years or so, we see NLP technologies being widely deployed, impacting billions of people directly and indirectly. Numerous instances have also surfaced where it is clear that adequate thought was not given to the development of those systems, leading to various adverse outcomes. It has also been well-established that a crucial ingredient in developing better systems is to engage with a wide ensemble of literature and ideas, especially bringing in ideas from outside of CS (say, psychology, social science, linguistics, etc.) Against this backdrop, the picture of NLP's striking lack of engagement with the wider research literature — especially when marginalized communities continue to face substantial risks from its technologies — is an indictment of the field of NLP. Not only are we not engaging with outside literature more than the average field or even just the same as other fields, our engagement is, in fact, markedly less. A trend that is only getting worse.

The good news is that this can change. It is a myth to believe that citation patterns are "meant to happen", or that individual researchers and teams do not have a choice in what they cite. We can actively choose what literature we engage with. We can work harder to tie our work with ideas in linguistics, psychology, social science, and beyond. We can work more on problems that matter to other fields, using language and computation. By keeping an eye on work in other fields, we can bring their ideas to new compelling forms in NLP.

[4]https://huggingface.co/spaces/jpwahle/field-diversity

## Limitations

As stated earlier, mechanisms of field-to-field influence are complex and subject to a variety of influences such as social factors (Cheng et al., 2023), centralization of the field on benchmarks and software (Gururaja et al., 2023), or publishing structures (Kim et al., 2020). This paper primarily makes use of citations to quantify influence which comes with many limitations. One such limitation is the lack of nuance that arises from counting citations; not all citations are equal—some cited work may have been much more influential to the citing work than others (Zhu et al., 2015). Further, citation patterns are subject to various biases (Mohammad, 2020a; Ioannidis et al., 2019; Nielsen and Andersen, 2021).

Although this current work relies on the assignment of hard field labels to papers, in reality, a paper may have a fuzzy membership to various fields. Work on automatically determining these fuzzy memberships will be interesting, but we leave that for future work. Similarly, there is no universally agreed ontology of fields of study, and different such ontologies exist that differ from each other in many ways. We have not done experiments to determine if our conclusions hold true across other ontologies. To counter these limitations, there exist works to study interdisciplinary and impact without bibliometrics (Saari, 2019; Carnot et al., 2020; Rao et al., 2023), and future work should explore applying these approaches to our questions.

For this work, we rely on the S2FOS classifier to categorize our 209m papers into research fields. Although S2FOS accuracy is at 86% for its 23 fields (e.g., medicine, biology), other alternatives such as S2AG (Kinney et al., 2023) can also be incorporated to obtain more accurate results. The first-level subfields attributed to the papers in our CS corpus use the CSO (Salatino et al., 2020) as a proxy, which was also applied in other domains (Ruas et al., 2022; Wahle et al., 2022). However, other alternatives such as the Web of Science, and SCOPUS can also be considered.

Additionally, recent work suggests adjusting for the general increase in publications (Kim et al., 2020) and the citational rate in each field (Althouse et al., 2009), finding that it affected their findings of citational diversity within fields and impact factors respectively. We do not expect this to change the conclusions we draw in this work, but it might be worth verifying.

Much of this work is on the papers from the *ACL Anthology (AA)*. It includes papers published in the family of ACL conferences as well as in other NLP conferences such as LREC and RANLP. However, there exist NLP papers outside of AA as well, e.g., in AI journals and regional conferences.

## Ethics Statement

As most of our experiments use the number of citations as a proxy to characterize research fields, some concerns should be discussed to avoid misinterpretation or misuse of our findings. The low number of ingoing or outgoing citations should not be used to justify diminishing a particular field or removing their investments. Because of our choice of citations as a primary metric, the other dimensions of our study, i.e., CFDI, influence, diversity, and insularity, are also subject to the same concerns. Decisions involving science should be evaluated through more than one aspect. Although still imperfect, a multi-faceted evaluation incorporating characteristics such as popularity, relevance, resources available, impact, location, and time could help to mitigate the problems of shallow analysis.

## Acknowledgements

This work was partially supported by the DAAD (German Academic Exchange Service) under grant no. 9187215, the Lower Saxony Ministry of Science and Culture, and the VW Foundation. Many thanks to Roland Kuhn, Andreas Stephan, Annika Schulte-Hürmann, and Tara Small for their thoughtful discussions.

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

| Category | Count ($\downarrow$) |
|---|---|
| Medicine | 45 693 661 |
| Biology | 22 899 911 |
| Computer Science | 15 102 871 |
| Chemistry | 15 014 729 |
| Engineering | 13 991 616 |
| Physics | 12 809 343 |
| Materials Science | 12 211 647 |
| Psychology | 9 594 542 |
| Environmental Science | 7 115 511 |
| Business | 7 096 738 |
| Education | 6 197 121 |
| Mathematics | 6 142 561 |
| Economics | 6 035 779 |
| Political Science | 5 578 326 |
| Agricultural And Food Sciences | 5 230 208 |
| Sociology | 4 094 792 |
| History | 3 669 722 |
| Art | 3 397 837 |
| Geology | 3 372 109 |
| Geography | 3 173 289 |
| Philosophy | 1 680 010 |
| Law | 944 678 |
| Linguistics | 880 043 |

Table 2: Number of papers per field for all 23 fields.

## A Appendix

### A.1 Demo for Citation Field Diversity

We have developed a freely accessible web-based demonstration to promote cognizance of disciplinary diversity in academic citations. Users can input any paper's Semantic Scholar ID, ACL Anthology link, and PDF file, and our system yields salient data concerning the interdisciplinary scope of the cited literature. One can also input author profiles or proceeding links from Semantic Scholar and ACL Anthology. The interface visualizes the distribution of the CFDI for all NLP papers published until 2022, juxtaposing this with the CFDI of the paper inputted by the user. Figure 17 in the Appendix shows an overview of the demo, which is available at

https://huggingface.co/spaces/jpwahle/
field-diversity.

### A.2 Supplementary Dataset Details

Table 2 shows the number of papers per field, showing that medicine, with 45.7m papers, is the largest field, followed by biology with 22.9m papers. CS is the third largest field, with 15m publications. Linguistics is the smallest one, with only 880k publications overall.

### A.3 Supplemental Experimental Results

In addition to the primary results presented in this paper, we describe supplementary results in the form of additional statistics and plots.

#### A.3.1 Results for NLP Subfields

We extended our analysis also for NLP subfields, with three additional questions answered in the following.

**SQ1.** *Which subfields within NLP have been the most frequently cited by works outside NLP? Which subfields within NLP have cited research from other fields the most?*

**Ans.** We categorize papers into subfields of NLP by computing the 200 most frequent bigrams of paper titles and manually assigning them to one of the 24 ACL Rolling Review (ARR) categories (e.g., ethics, generation). For example, the paper titled "Large Language Models in Machine Translation" (Brants et al., 2007) was assigned to the NLP subfield "machine translation". We add an additional category for shared tasks (e.g., SemEval tasks). We then measure the number of citations between subfields and other research fields.

*Results.* Figure 9 shows the percentage of citations from an NLP subfield to another CS subfield (a) or a non-CS research field (b). The denominator for (a) is all CS citations and for (b) non-CS citations.

**9 (b)** Linguistics plays a significant role in non-CS, influencing lexical semantics (32%) and machine translation (30%). Often surpassing other non-CS fields, math is cited most by ML for NLP (35%) and machine translation (32%) over all non-CS fields. Psychology also stands out, impacting NLP

applications (19%), lexical semantics (16%), and sentiment analysis (16%).

**9 (a)** Shifting the focus to CS disciplines, AI' and ML have a substantial influence on the subfield of lexical semantics, with AI' contributing 32% and ML 16% of the influence. For sentiment analysis, a broader mix of disciplines comes into play, with AI' (24%) and data mining (16%) leading the charge, followed by information retrieval (IR) and ML, each contributing 12%.

*Discussion.* The results indicate that machine translation, ML for NLP, and lexical semantics exhibit substantial influences from various disciplines, such as AI, mathematics, and linguistics. NLP applications have a broad influence, particularly in psychology and medicine, spanning both CS and non-CS domains.

When connecting these subfield findings to the general trends in NLP, we can see the broader picture of the field's interdisciplinary nature and evolution. The influence of linguistics and sociology has decreased over time, aligning with the overall shift in NLP towards more quantitative and empirical methodologies. On the other hand, the exponential increase in the influence of mathematics and the growing significance of psychology is reflected even more strongly in specific subfields.

**SQ2.** *How diversely specific subfields of NLP cite other research fields?*

**Ans.** We measure the average CFDI for NLP.

*Results.* The horizontal black line at zero in Figure 10 shows the average outgoing CFDI for NLP per year. While the overall citational field diversity NLP has been declining over time, as discussed in Q3, specific subfields deviate from that downward trend (i.e., they even decline more or they do not decline as much). Machine translation and ML for NLP started at average CFDI (considering all documented fields) in 1990 but rapidly decreased to 0.14 (machine translation) and 0.06 (ML for NLP) in 2010. ML for NLP came back close to average NLP diversity in 2022 (-3.4%), but machine translation has remained at -0.10 diversity compared to the average. Dialogue and interactive systems started with much higher-than-average diversity in 1996 (+0.11) but dropped below average in 2018 and 2022 (-0.05). Multilinguality and language diversity are clear winners, with consistently more than a +0.10 average diversity index. It is worth noting that multilinguality and language diversity

were only recently recorded without subfield assignments since the 2000s.

*Discussion.* The consistently high diversity in multilinguality and lexical semantics suggests a broad interdisciplinary engagement, reflecting these areas' inherent complexity and wide-ranging applications. The declining diversity in machine translation, dialogue, and interactive systems could indicate a consolidation of research within specific methodologies or theories or a focus on refining existing techniques rather than integrating new perspectives. The recent resurgence in field diversity within machine learning for NLP is encouraging, as it may signal a renewed openness to cross-disciplinary influences and innovative approaches.

**SQ3.** *To what extent do papers in NLP subfields cite themselves? How does that score compare between subfields?*

**Ans.** We measure the percentage of intra-field citations, i.e., the number of citations from an NLP subfield to itself over the total number of citations from that subfield. A high intra-field citation percentage means a field is relying predominantly on works from within, while low-self citations indicate a high amount of reliance on works outside that field

*Results.* Figure 11 shows the intra-field citation percentage of NLP subfields. Machine translation has the largest amount with 65.7% while summarization and sentiment analysis are at 49.6% and 47.4%, respectively. Shared tasks, applications, or ethics, have much lower ratios of 22.7%, 16.5%, and 6.1%, respectively.

*Discussion.* Some fields, such as machine translation, have grown strongly over time. Therefore, more relevant works are published in a growing community, and more researchers are keen to cite work from their community. However, fields of comparable sizes, such as ML for NLP, have a much lower intra-field citation percentage. Therefore, fields like machine translation should be conscious about whether they are engaging with enough relevant work from fields outside of their own niche.

### A.3.2 Limitations of NLP Subfield Analysis

We limited our categorization to subfields to bigrams of NLP paper titles (~80k). While in many cases, such assignments are precise, there are some caveats. We also did not account for any other n-grams in the paper's text, as the coverage of all

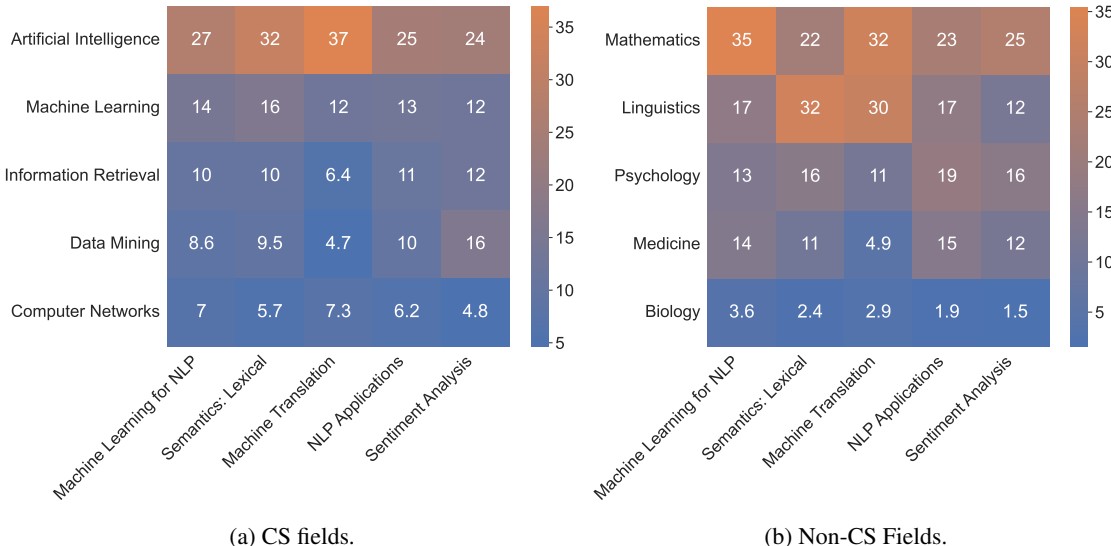

(a) CS fields.                    (b) Non-CS Fields.

Figure 9: The percentage of citations a subfield cites (a) CS subfields and (b) non-CS fields for the five most cited fields and subfields, respectively. Percentages are using (a) citations to CS and (b) overall citations to non-CS fields as denominators.

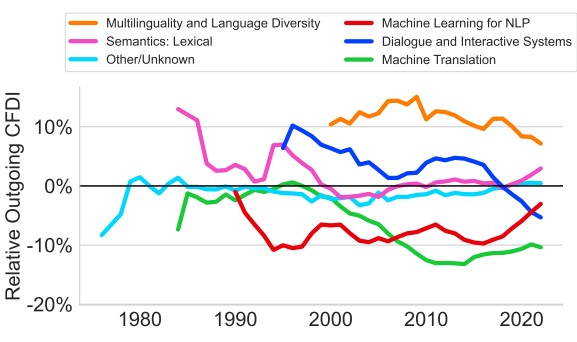

Figure 10: The outgoing CFDI of the six largest sub-fields relative to average annual NLP diversity.

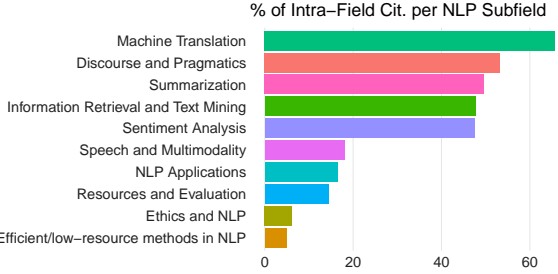

Figure 11: The percentage of intra-field citations of the five most self-citing and five least self-citing NLP subfields.

possible combinations would be restrictive. Additionally, the mapping of these bi-grams to one of the categories in the ACL Rolling Review (ARR) only considers the instructions of the 2023 ACL reviewer form which can change over time. We assume the instructions provided for 2023 result from an extensive discussion and curation process between the conference organizers over the years.

### A.3.3 Diachronic Trends of CS Subfields

In the following, we provide additional diachronic analysis of Q1 for CS subfields. Figure 12 shows the citation percentage of the three most prominent, and two additionally selected CS subfields with steep increases or declines in citation percentage. AI' has consistently the most influence on NLP (range: 21.9%–27.5% outgoing CS citations). In the 1980s, there has been a substantial rise in

citations to ML (3.9% to 10.1% outgoing CS citations). Citations to IR relevance decreased during the 1980s, but in the 1990s, it returned to previous all-time highs.

### A.3.4 Comparison of Field Citations Overall

Figure 13 (a) shows the percentage of outgoing citations from NLP to non-CS fields and CS subfields with the same denominator, i.e., all outgoing citations. Using this plot, we can visualize how many citations linguistics receives from NLP in comparison to ML. Similarly, Figure 13 (b) shows the percentage of incoming citations from NLP to non-CS fields and CS subfields in the same scale.

### A.3.5 Extended Results on Relative Citation Prominence

Figure 14 shows the ORCP from Q3 for NLP, CS, linguistics, math, and psychology as well as the

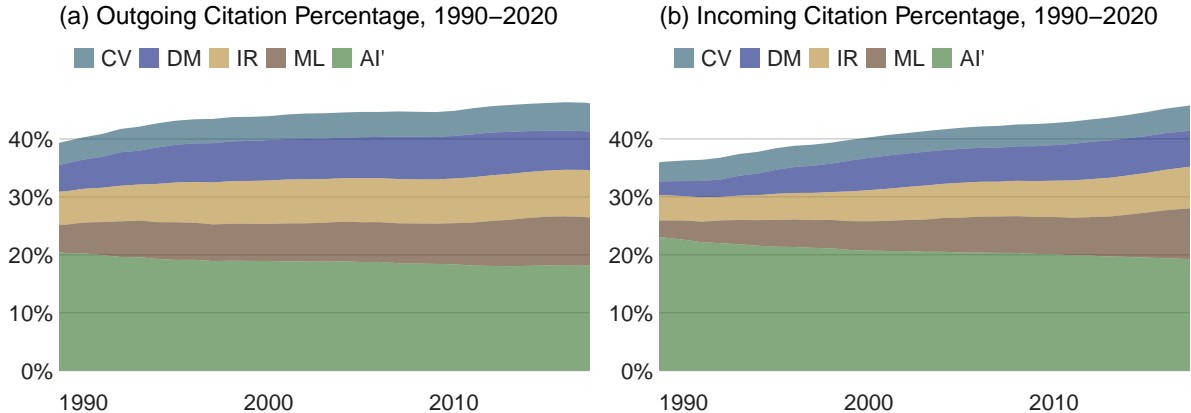

Figure 12: The percentage of citations from (a) NLP to CS subfields and (b) CS subfields to NLP in relation to all CS citations from and to NLP. CV - Computer Vision; DM - Data Mining; IR - Information Retrieval; ML - Machine Learning; AI - Artificial Intelligence.

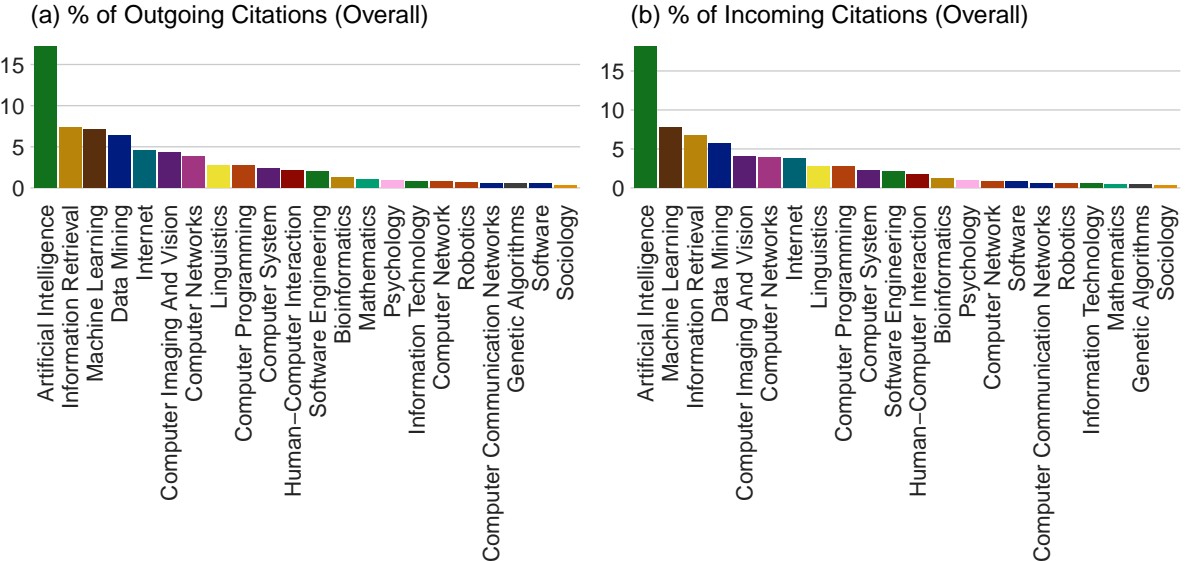

Figure 13: The overall percentages of (a) outgoing citations from each field over all citations from NLP and (b) of incoming citations from each field over all citations to NLP.

range of minimum and maximum ORCP over all 23 general fields of study. Figure 16 and Figure 15 show the respective IRCP plots.

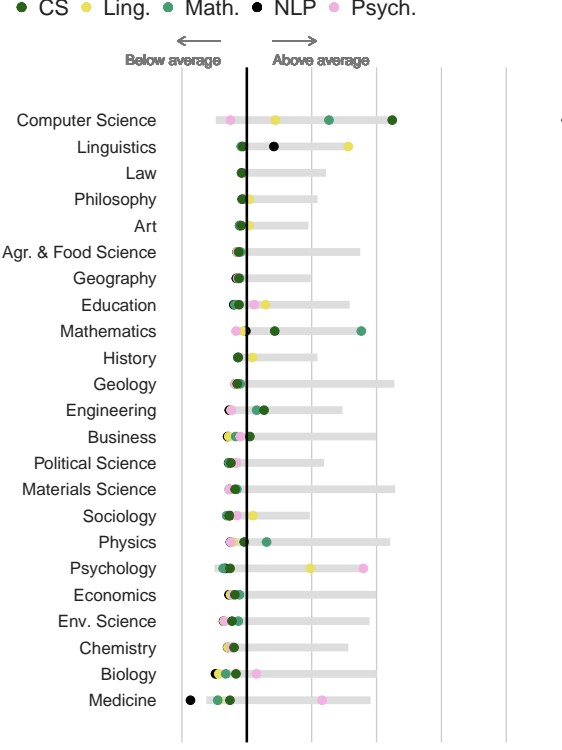

Figure 14: Outgoing Relative Citational Prominence (ORCP) scores between NLP, CS, linguistics, math, and psychology and all 23 fields of study.

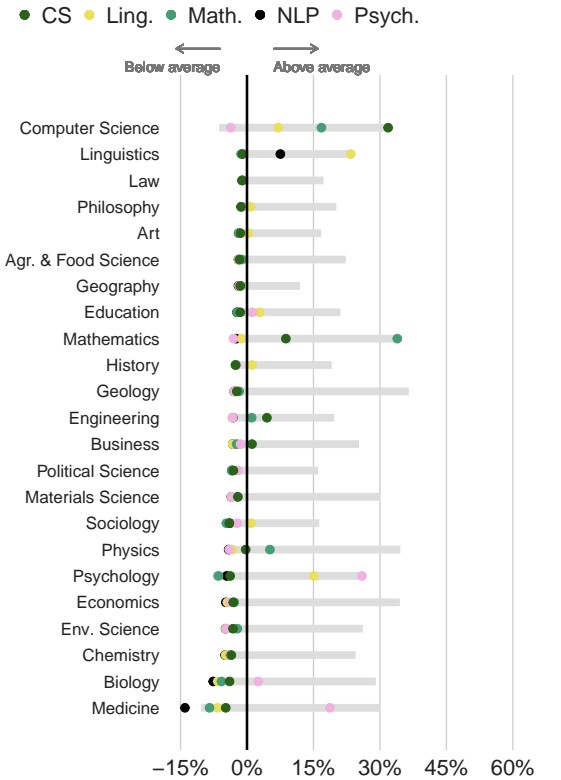

Figure 15: Incoming Relative Citational Prominence (IRCP) scores between NLP, CS, linguistics, math, and psychology and all 23 fields of study.

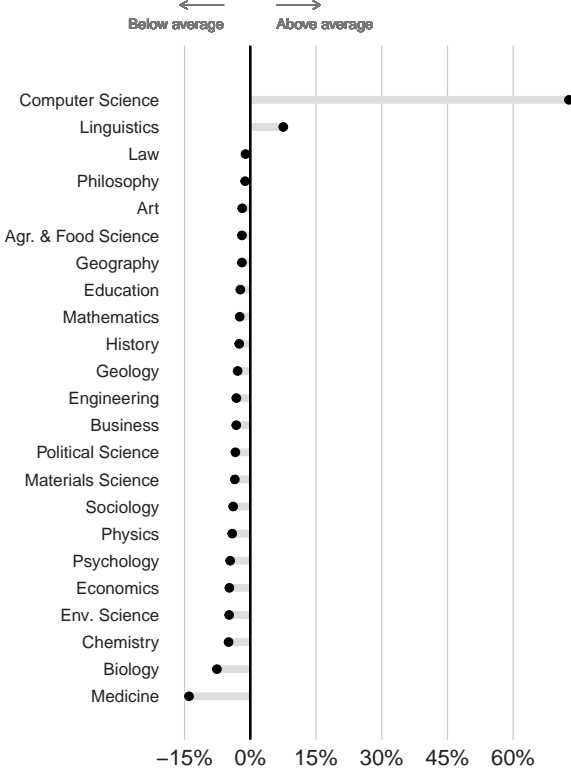

Figure 16: NLP's Incoming Relative Citational Prominence (IRCP) scores for 23 fields of study.

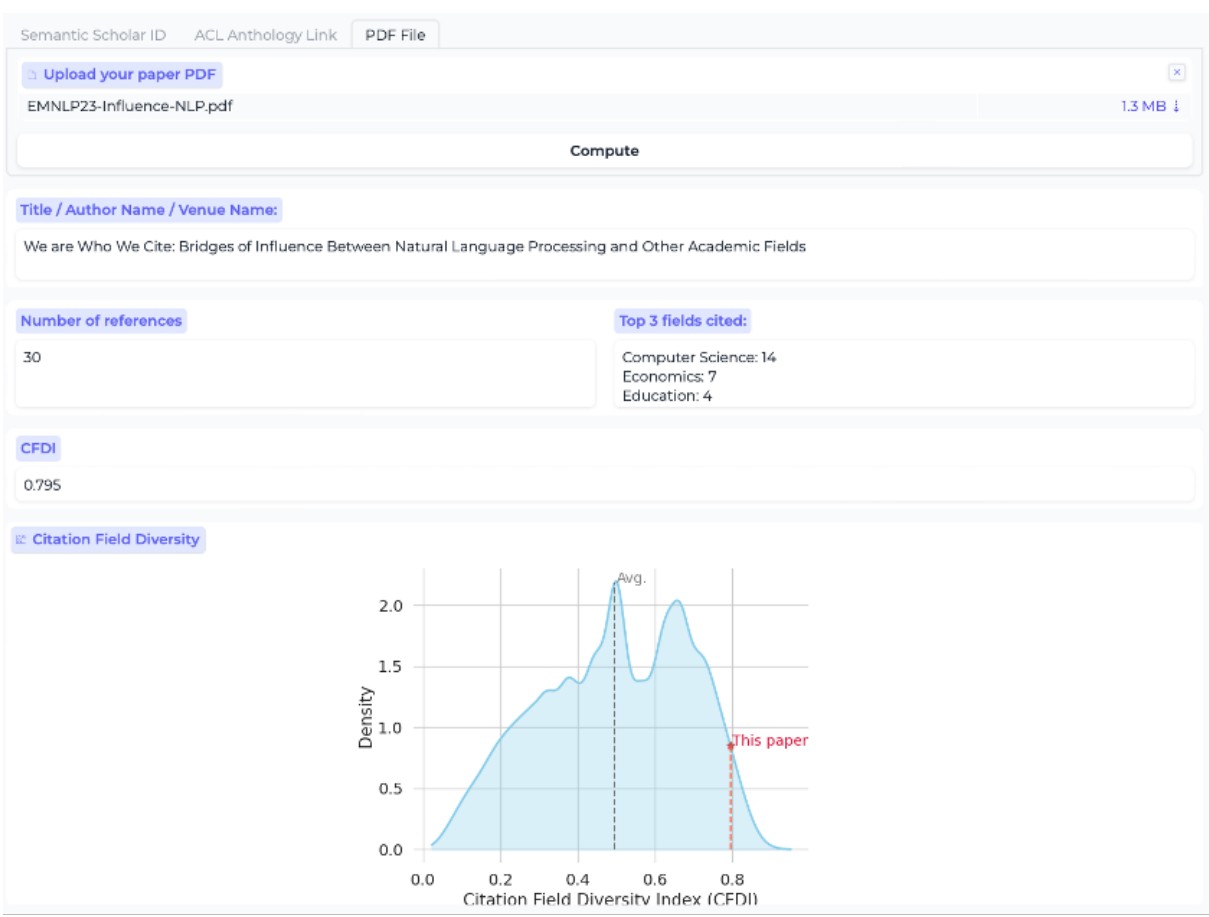

Figure 17: A web demo to compute field diversity metrics for a paper, author, or proceeding.