# OpenReview forum: "We are Who We Cite: Bridges of Influence Between Natural Language Processing and Other Academic Fields"
_EMNLP/2023/Conference — EMNLP 2023 Main_

### Official Review · Reviewer_H61G · 2023-08-03

**Soundness:** 4

**Excitement:**

4: Strong: This paper deepens the understanding of some phenomenon or lowers the barriers to an existing research direction.

**Missing References:**

Not that it is imperative to cite them, but the following are examples that analyse (or try to predict) research trends in different areas, generally focusing on used text, which might be worth looking at:

Dietz, L.; Bickel, S.; Scheffer, T. (2007). Unsupervised prediction of citation influences. ACM Int. Conf. Proc. Ser. 2007, 227, 233–240.

Francopoulo, G.; Mariani, J.; Paroubek, P. (2016). Predictive modeling: Guessing the NLP terms of tomorrow. In Proceedings of the 10th International Conference on Language Resources and Evaluation (LREC 2016), Portorož, Slovenia, 23–28 May 2016; pp. 336–343.

Saari, E. (2019). Trend Analysis in AI Research. Ph.D. Thesis, Tampere University, Tampere, Finland

Carnot, M. L., Bernardino, J., Laranjeiro, N., & Gonçalo Oliveira, H. (2020). Applying text analytics for studying research trends in dependability. Entropy, 22(11), 1303.

**Paper Topic And Main Contributions:**

This is a study on Natural Language Processing (NLP) research trends with a special focus on academic fields typically cited by or citing NLP papers.
It is based on metadata from the ACL Anthology and Semantic Scholar, and provides very interesting insights on the trends, overall and its evolution since 1980, which really makes us think.

The paper is structured on several questions and their answers, illustrated with nice plots. However, in most cases, the answers are limited to describing the contents of plot, with very limited discussion (Q6 and Q7 are an exception) and extrapolated conclusions. It would be interesting to go deeper on the possible causes of some observed trends, even if speculating, to some extent.

Considering the previous limitation as well as the plots left in the Appendix, I wonder whether this paper is better suited for a journal than for a conference.

**Questions For The Authors:**

Question A: The method for collecting citations, especially from other fields, is not completely clear. Are they accessible from Semantic Scholar? The actual process for getting them should be provided.

Question B: Why are some plots ending in 2020 and not 2022?

Question C: Would it be possible to correlate some findings with actual works or trends? For instance, any idea for the cause of the 2006(?) peak in the plot of Figure 2a? Or for the start of the decrease of outgoing linguistics citations in the plot of Figure 3a?

Question D: As mentioned, arXiv and related services seem to have an impact on Q7 (Do well-cited NLP papers have higher citational field diversity?), possibly hiding the real picture. I would add the fact that NLP has become more applicational / it is very easy to use state-fo-the-art tools out-of-the-box, so researchers that use these tools tend not to cite them or end up just including their URL.
Would it be possible to speculate on the trend after 2022, when virtually every researcher heard of NLP and the usage of Large Language Models exploded?

**Reasons To Accept:**

A paper with the potential of raising very interesting discussions.

Useful insights on NLP and its connection to other academic fields, illustrated by very nice graphics.

Results are accessible through a web application, which further provides visualisations for given papers.

**Reasons To Reject:**

Discussions are very limited.

Related Work is not focused on previous studies on research trends for specific areas, their methodologies and adopted metrics.

**Reproducibility:**

3: Could reproduce the results with some difficulty. The settings of parameters are underspecified or subjectively determined; the training/evaluation data are not widely available.

**Reviewer Confidence:**

4: Quite sure. I tried to check the important points carefully. It's unlikely, though conceivable, that I missed something that should affect my ratings.

**Typos Grammar Style And Presentation Improvements:**

Why NLP -> Why Natural Language Processing (NLP)
focus on Natural Language Processing (NLP) -> focus on NLP

The Introduction should end with a description of the paper structure.

Part of the Related Work is not really "related work". For instance, given that the study does not focus specific publications, are the citations to examples of "synergistic interaction" of multiple disciplines really necessary?

Some acronyms like IR and CV are not introduced in the body of the paper (only in the Appendix).

---

> ### Author Rebuttal · Authors · 2023-08-28
>
> Thank you for your thoughtful review and for finding the work to bring out interesting insights on the influence between NLP and other academic fields. We also thank you for highlighting its "potential of raising very interesting discussions" within the academic community.
>
> Question A: Re: method for collecting citations
>
> Answer A: The citation information (which paper cites which paper, year of publication of a paper, etc.) is accessible through SemanticScholar’s data dump, which is regularly updated and openly available. We will provide the source code on GitHub. We will also update the paper with additional details to clarify the data processing so that the work can be easily reproduced.
>
> Question B: Why are some plots ending in 2020 and not 2022?
>
> Answer B: Citation-based plots generally use 2020 as the limit because papers published within the last three years didn’t have enough time to accumulate citations—the number of citations a paper receives in AA peaks after 3 years. Plots not affected by the absolute citation count of papers (e.g., Fig 6: Avg. number of fields per paper) also use the most recent papers until 2022. We will make this clearer in the main body of the paper.
>
> Question C: Would it be possible to correlate some findings with actual works or trends? For instance, any idea for the cause of the 2006(?) peak in the plot of Figure 2a? Or for the start of the decrease of outgoing linguistics citations in the plot of Figure 3a?
>
> Answer C: We enjoyed the idea of pinpointing specific works to trends in our study. For example, one could measure which highly cited papers have led to much more/less linguistics citations over time. We will include that thought for our future work.
>
> Question D: NLP has become more applicational / it is very easy to use state-fo-the-art tools out-of-the-box, so researchers that use these tools tend not to cite them or end up just including their URL. Would it be possible to speculate on the trend after 2022, when virtually every researcher heard of NLP and the usage of Large Language Models exploded?
>
> Answer D: We think tracking citations of other fields to NLP is especially interesting going forward. Given the tremendous and widespread impact of LLMs, we expect a much higher amount of citations and a much wider set of fields using and citing NLP technologies. As you point out, perhaps *some* of these works will use NLP technologies but not cite NLP papers. However, we cannot predict the extent to which this will happen. Our guess though is that we will still see a marked rise in citations to NLP papers from other fields. In that regard, the results in this paper act as a great comparison point – and we hope future work will revisit the questions raised here for papers published between 2022 and a later point in time.
>
> Once again, thank you for your time and for engaging with the material in this paper and our response. We see the time and thought you have put into this and very much appreciate it.

---

### Official Review · Reviewer_GqaD · 2023-08-04

**Soundness:** 3

**Excitement:**

3: Ambivalent: It has merits (e.g., it reports state-of-the-art results, the idea is nice), but there are key weaknesses (e.g., it describes incremental work), and it can significantly benefit from another round of revision. However, I won't object to accepting it if my co-reviewers champion it.

**Paper Topic And Main Contributions:**

The paper analyses citations of/in NLP papers to assess the influence of NLP research on/from other fields, including a diachronic analysis.
To conduct such an investigation, authors propose a dataset of metadata associated with ∼77k NLP papers, ∼3.1m citations from NLP papers to other papers, and ∼1.8m citations from other papers to NLP papers. Notably, the metadata includes the field of study and year of publication of the papers cited by the NLP papers, the field of study and year of publication of papers citing NLP papers, and the NLP subfields relevant to each NLP paper.


**Reasons To Accept:**

The paper is well-written and the research questions interesting, even though the results somehow expected, mainly for some aspects (e.g., the domination of computer science citations).

**Reasons To Reject:**

As stated also by the authors in the Limitations Section, the paper primarily makes use of citations to quantify influence.

**Reproducibility:**

3: Could reproduce the results with some difficulty. The settings of parameters are underspecified or subjectively determined; the training/evaluation data are not widely available.

**Reviewer Confidence:**

3: Pretty sure, but there's a chance I missed something. Although I have a good feel for this area in general, I did not carefully check the paper's details, e.g., the math, experimental design, or novelty.

---

> ### Author Rebuttal · Authors · 2023-08-28
>
> Thank you for your review and for finding the research questions raised to be interesting and the paper to be well-written.
>
> Regarding the use of citations to measure influence:
>
> This study aimed to measure **citational influence** between fields. This is because even though one field can influence another field in various ways, the amount of citations to a field is a direct and substantive indicator of influence. In fact, it is hard even to imagine what else one can use that will be as effective in determining influence other than citations. Engagement with literature, researchers, or scientific events from other fields eventually leads to citations from one work to another. Of course, we welcome ideas for other measurable markers of influence that we can add (in addition to looking at citations).

---

### Official Review · Reviewer_k77Z · 2023-08-05

**Soundness:** 4

**Excitement:**

4: Strong: This paper deepens the understanding of some phenomenon or lowers the barriers to an existing research direction.

**Missing References:**

* A recent (contemporaneous) paper that may be of interest for citing community opinion: _What do NLP researchers believe? Results of the NLP community metasurvey_ (Michael et al., ACL 2023).

* Assorted work on citation networks from Jevin West's group may be relevant, e.g. _The influence of changing marginals on measures of inequality in scholarly citations: Evidence of bias and a resampling correction_ (Kim et al., Sociological Science 2020), mentioned above; _Bursting scientific filter bubbles: Boosting innovation via novel author discovery_ (Portenoy et al., CHI 2022), which introduces a system to enable discovery of other researchers outside one's "bubble".

**Paper Topic And Main Contributions:**

This meta paper examines influence between NLP and other fields through the lens of citation frequency, and in doing so calls for increased engagement with other fields as an important aspect of responsible NLP research. Contributions include:

* Publication, citation, and field of study data compiled from the ACL Anthology and Semantic Scholar. (The paper also indicates that a tool for exploring this data will be made available upon acceptance.)
* Suggested metrics for measuring in- and out- direction citations for a field relative to others (incoming/outgoing relative citational prominence) and citation field diversity.
* Analysis using said metrics to address a number of research questions: e.g., diversity of NLP citations to itself vs. other fields (CS or not), citation frequency by papers in other fields to NLP, changes in citation patterns over time, and other related effects.
* Results demonstrate increased insularity of NLP relative to other fields.

**Questions For The Authors:**

Thank you for an interesting read -- lots to engage with! Some questions:

* There is past work indicating that several measures used for assessing citation inequality, including the Gini coefficient, may need correction to account for paper/citation increases over time (_The influence of changing marginals on measures of inequality in scholarly citations: Evidence of bias and a resampling correction_, Kim et al., Sociological Science 2020). That scenario differs slightly as the measurement is of citation inequality across papers, not fields; is a similar correction necessary for your Gini-based measure at the field level (CFDI)?

* The x-axis ranges in charts are a bit inconsistent -- Fig. 2 & 3 are 1990-2020, Fig. 5 is 1980-2020, and Fig. 6 & 7 are 1980-2022 -- is there a particular reason for doing so? For other charts aggregating time away (e.g., Fig. 1 and 4), is the data used from the full 1965-2022 range or a similar subset as the other figures?

* For the field information in Semantic Scholar & CSO, do you have a sense of whether accuracy may degrade for papers that are interdisciplinary / highly unusual to a field?

* Not a question, but please feel free to engage with other points above and/or note if I've misunderstood/misrepresented anything from your paper in this review.

**Reasons To Accept:**

* The paper offers valuable meta-analysis of NLP in- and out-citation trends: NLP is a fundamentally interdisciplinary field that has broad applications due to the ubiquity of language, and having a better understanding of research influence in both directions can help increase awareness of potential insularity.

* No lack of substance in the paper -- the analysis goes in depth examining in- and out-bound citation percentages (absolute and in relation to norms across fields), diversity in cited fields based on the Gini index, whether well-cited papers tend to have high citational field diversity, and in the appendix additional results on subfields of NLP. (The 8 page limit is a major constraint here.)

* The writing is generally clear and the paper is well-organized.

**Reasons To Reject:**

I think most of my concerns with this paper stem from the "position paper" framing, mostly in the introduction and conclusion, that asserts interdisciplinary influence from other fields to NLP and vice versa (as a function of citation diversity) as ethical imperative. I don't personally disagree with that assertion, but the analysis presented in section 3 feels somewhat disjoint from that:

* I didn't fully understand why the emphasis between CS and non-CS in considering interdisciplinarity -- my instinct would be toward focusing on AI/ML vs. not (if NLP has simply been subsumed into ML to the detriment of everything else) or something like quantitative/statistical vs. not (if NLP has become overwhelmingly quantitative, with loss of linguistics/psychology/other grounding). Given the motivation to better account for social risks/implications, it feels odd to "penalize" areas of CS that actually align tightly with those goals (e.g., HCI and security). I would love to see a bit more nuance in that aspect even if the results don't significantly change.

* Similarly, NLP citing certain fields less frequently compared to the average across all fields is itself not necessarily a concern -- e.g., for more distant fields (e.g., agriculture & food science, materials science, and geography), it is not unreasonable that NLP cites them less frequently than other fields do (Figure 4) -- a bit more nuance in why fields may be cited (methods, substantive grounding, ethical concerns, etc.) may help clarify when lower-than-average / decreased citation proportion is indeed concerning.

* I felt that the rhetoric in the introduction and discussion/conclusion verged towards assumption of moral judgement and/or popular opinion that detracted from the substance of the paper. (Tone is of course subjective; though the argumentation could be refined further, I don't think this itself is a strong reason for rejection.)

  - Arguments are sometimes attributed to community opinion in ways that should warrant either specific citation or reframing to come from the authors (all italics mine): "While _some have argued_ that CS dominance has become extreme (and harmful)..." (L124); "_There have been concerns in the academic community_ that early-stage researchers often categorize papers not from their immediate field of study as irrelevant... _researchers feel_ that it is difficult to keep track of core NLP papers itself" (L536).

  - The paper ends with: "this striking lack of engagement with the wider research literature, especially when marginalized communities face substantial risks, is an indictment of the field of NLP" (L590). The results certainly support trends towards increased citation insularity of NLP (alone and in comparison to other fields), but it is a tall jump from that to moral indictment. (A devil's advocate question: in this measurement setup, what might we deem to be sufficient interdisciplinary citation to minimize such hazards? e.g., constant ratio over time, similar ratio to other fields, etc.)

_edit/clarification: please interpret the length of this section more as "this paper was engaging and I had some contrary thoughts", not "these are major reasons to reject."_

**Reproducibility:**

4: Could mostly reproduce the results, but there may be some variation because of sample variance or minor variations in their interpretation of the protocol or method.

**Reviewer Confidence:**

4: Quite sure. I tried to check the important points carefully. It's unlikely, though conceivable, that I missed something that should affect my ratings.

**Typos Grammar Style And Presentation Improvements:**

Minor suggestions/errors:

* Some words are unnecessarily capitalized (e.g., "language processing", "less", "linguistics" in the abstract).
* Table 1 hyphenates in-citations and out-citations; perhaps consider doing the same for in-cites and out-cites more broadly (incites is a very different word! got a bit tripped up every time I saw it).
* Fig. 14 & 15: the colors for math & CS read as basically identical to me.

---

> ### Author Rebuttal · Authors · 2023-08-28
>
> We are grateful to the reviewer for their detailed and insightful review.
>
> Re it feels odd to "penalize" areas of CS that actually align tightly with those goals:
>
> We are not sure whether we got the main point here. We are not penalizing any particular field; instead, the goal was to provide information on who NLP cites and who cites NLP through multiple ways of aggregation (from and to various fields, from and to AI, etc.).
>
> Re NLP citing certain fields less frequently compared to the average across all fields is itself not necessarily a concern:
>
> Yes, we agree. It is not a concern on its own that NLP cites specific fields more/less than others. This work provides the information, and it is up to the community or individual researchers to reflect on it for themselves. We will make this clearer in the text.
>
> Re rhetoric in the introduction and discussion/conclusion:
>
> This is indeed tricky, and so we appreciate the opportunity to discuss this with you; especially given your thoughtful review and substantive engagement with our submission. Our line of thought is as follows.
> NLP technologies are being widely deployed, and numerous cases have surfaced where it is clear that adequate thought was not given before developing those systems, leading to harm, especially to marginalized communities. It has also been well-established that a crucial ingredient in developing better systems is to engage with literature and bring in team members from outside of CS (say, psychology, social science, linguistics, etc.)
> Given this context, we think it is particularly damning for us as a community that not only are we not engaging with outside literature more than the average field or even just the same as other fields, we are in fact, doing so markedly less! And this trend is only getting worse. We will have an extra page if the paper is accepted, and we can make this point more clearly with an explicit “Discussion” section. It is better for us as a community to look critically at ourselves rather than outsiders raise critical issues.
>
> Regarding the engagement discussion in introduction:
>
> We do not mean to exert moral judgment. However, our goal is to convey that it is a myth to think that citation patterns that have occurred were meant to happen, and we did not have a choice in that. As a scientific community, we should be willing to take responsibility for what literature we choose to engage with and what we choose to ignore. And if we believe that, then it is important to know about the broad trends in what literature we engage with and who we cite.
>
> Question A: Re corrections to Gini for scale.
>
> Answer A: Thank you for bringing this to our attention. Our study measures inequality as a “per-paper” metric. We measure how unequal a single paper cites different fields in their references. Our aggregator over all papers then becomes the mean, which, in its limits, is not affected by the amount of papers. Indeed, for incoming citations to a paper, our metric could be mildly affected, and we are exploring suitable methods to correct for this impact. However, we do not expect it to change any of the broad conclusions of our work, especially since we are interested in comparisons across fields.
>
> Question B: Re x-axis ranges.
>
> Answer B: Aggregated plots use the entire time range. Citation-based plots generally use 2020 as the limit because papers published within the last three years didn’t have enough time to accumulate citations—the number of citations a paper receives in AA peaks after 3 years. Plots not affected by the absolute citation count of papers (e.g., Fig 6: Avg. number of fields per paper) also use the most recent papers until 2022. We will add a comment on the difference in dates prominently in the main body of the paper.
>
> Question C: For the field information in Semantic Scholar & CSO, do you have a sense of whether accuracy may degrade for papers that are interdisciplinary / highly unusual to a field?
>
> Answer C: Semantic Scholar and CSO provide results on their classification as aggregated results, not individually for multi-field papers. The variation in detection results could be higher because the domain shift is less likely to fall in the same distribution as the original field. From our non-systematic manual exploration of results, we found only very few cases in which the multi-field assignment was incorrect.
>
> Once again, thank you for your time and for engaging with the material in this paper and our response. We see the time and thought you have put into this and very much appreciate it.

---

### Meta-Review · Area_Chair_t3wp · 2023-09-16

**Recommendation:** 5

**Metareview:**

This manuscript examines how deeply NLP research engages with other fields as approximated by citations outside of NLP venues. The manuscript provides in-depth analysis of some of their findings.

In general, reviewers find many reasons to accept this manuscript and only very few reasons not to, some of which are an excellent basis for further work investigating this. Indeed, the limitation section covers many of the limitations identified by the reviewers.
The most compelling reason for rejecting the manuscript is that certain analyses sections are primarily engaged with describing the figures rather than discussing the results themselves. However, this seems very easily addressable by the CR deadline.

Moreover, it's unclear to me why the authors rely on classification, when publication venues (e.g., ACM Transactions on Asian and Low-Resource Language Information Processing or GeoJournal) often explicitly name the field in the name. This information should be available from both citations and the publication venue of different papers. Using a classifier seems like an unnecessary way to introduce error into the analysis.

---

### Decision · Program_Chairs · 2023-10-07

**Decision:**

Accept-Main

**Comment:**

This manuscript examines how deeply NLP research engages with other fields as approximated by citations outside of NLP venues. The manuscript provides in-depth analysis of some of their findings.

In general, reviewers find many reasons to accept this manuscript and only very few reasons not to, some of which are an excellent basis for further work investigating this. Indeed, the limitation section covers many of the limitations identified by the reviewers.
The most compelling reason for rejecting the manuscript is that certain analyses sections are primarily engaged with describing the figures rather than discussing the results themselves. However, this seems very easily addressable by the CR deadline.

Moreover, it's unclear to me why the authors rely on classification, when publication venues (e.g., ACM Transactions on Asian and Low-Resource Language Information Processing or GeoJournal) often explicitly name the field in the name. This information should be available from both citations and the publication venue of different papers. Using a classifier seems like an unnecessary way to introduce error into the analysis.